# Essential Hypertension and Oxidative Stress: Novel Future Perspectives

**DOI:** 10.3390/ijms232214489

**Published:** 2022-11-21

**Authors:** Caterina Franco, Edoardo Sciatti, Gaia Favero, Francesca Bonomini, Enrico Vizzardi, Rita Rezzani

**Affiliations:** 1Division of Anatomy and Physiopathology, Department of Clinical and Experimental Sciences, University of Brescia, 25123 Brescia, Italy; 2Cardiology Unit 1, ASST Papa Giovanni XXIII, 24127 Bergamo, Italy; 3Interdepartmental University Center of Research “Adaption and Regeneration of Tissues and Organs-(ARTO)”, University of Brescia, 25123 Brescia, Italy; 4Italian Society of Orofacial Pain (SISDO), 25123 Brescia, Italy; 5Section of Cardiovascular Diseases, Department of Medical and Surgical Specialties, Radiological Sciences and Public Health, University of Brescia, 25123 Brescia, Italy

**Keywords:** essential hypertension, oxidative stress, melatonin, total antioxidant capacity, peripheral arterial tonometry, pulse wave velocity, systolic and diastolic blood pressure, cardiovascular risk

## Abstract

Among cardiovascular diseases, hypertension is one of the main risk factors predisposing to fatal complications. Oxidative stress and chronic inflammation have been identified as potentially responsible for the development of endothelial damage and vascular stiffness, two of the primum movens of hypertension and cardiovascular diseases. Based on these data, we conducted an open-label randomized study, first, to evaluate the endothelial damage and vascular stiffness in hypertense patients; second, to test the effect of supplementation with a physiological antioxidant (melatonin 1 mg/day for 1 year) in patients with essential hypertension vs. hypertensive controls. Twenty-three patients of either gender were enrolled and randomized 1:1 in two groups (control and supplemented group). The plasmatic total antioxidant capacity (as a marker of oxidative stress), blood pressure, arterial stiffness, and peripheral endothelial function were evaluated at the beginning of the study and after 1 year in both groups. Our results showed that arterial stiffness improved significantly (*p* = 0.022) in supplemented patients. The endothelial function increased too, even if not significantly (*p* = 0.688), after 1 year of melatonin administration. Moreover, the supplemented group showed a significative reduction in TAC levels (*p* = 0.041) correlated with the improvement of arterial stiffness. These data suggest that melatonin may play an important role in reducing the serum levels of TAC and, consequently, in improving arterial stiffness.

## 1. Introduction

Arterial hypertension is one of the most frequent cardiovascular disorders, more predominant in the older generation [1]. The Global Burden of Disease study defined hypertension as a key factor for disability-adjusted life years worldwide; moreover, it affects 1.13 billion people around the world (~20% female and ~24% male) [2,3] and is known for being one of the major causes of death and disability worldwide, leading to 9.4 million deaths annually [4,5,6]. 

According to the WHO’s ratings, overall, cardiovascular diseases (CVDs) accounted for 17.9 million deaths in 2019. At present, although the mortality rate has reduced, the prevalence of CVDs remains high, and it is important to note that hypertension is—among possible causes and risk factors for CVDs—one of the most easily preventable and controllable diseases [6,7]. Moreover, it is also one of the most significant comorbidities contributing to the development of stroke, myocardial infarction, heart, and renal failure [8], leading to early disability and dependency, reducing the life expectancy in patients, and increasing the cost of care [9,10,11]. 

Among all the cases of hypertension, approximately 95% of them are identified as essential hypertension (EH), a complex multifactorial haemodynamic and structural disorder characterized by an overall increased systolic blood pressure (SBP) of ≥140 mmHg and/or diastolic blood pressure (DBP) of ≥90 mmHg [12,13], resulting from the interplay of non-specific environmental and genetic factors [6]. Moreover, there are conditions that, even during pregnancy, could contribute to the development of early EH [14]. In recent years, many studies have focused on this topic, giving life to the now called “developmental origins of health and disease (DOHaD)”.

Physiologically, blood pressure (BP) values remain within a physiological range thanks to a fine balance maintained between factors that could increase BP and those that normalize it, working as compensatory mechanisms. It is when this slender and fragile dynamic stability fails that EH “appears” [15,16]. EH physiopathology surpasses the increase in BP; in fact, an important role is played by the delicate balance between vasodilators and vasoconstrictors [6]. When this balance is disturbed, it leads to endothelial dysfunction with an excessive release of vasoconstrictor substances [6,17,18]. In a second moment this alteration could lead to structural and functional changes in both the macro- and micro-structure of the vasculature [19] that can sometimes result in complete ventricular remodelling.

Next to the endothelial dysfunction, arterial stiffening must also be considered in EH etiopathogenesis: it is one of the main responses to chronical stress exerted by dyslipidaemia, ageing, and high BP itself. The arterial modulation to external stress limits the buffering capacity of the elastic arteries, exposes the microvasculature to increased pulsatile stress, and contributes to targeted organ damage via capillary rarefaction and microvascular ischemia [19,20], defining a general worsening of the overall clinical outcome.

Another important background mechanism in EH development [21] is represented by the increased oxidative stress and alterations in the total antioxidant capacity (TAC), which is a typical feature of many other CVDs [22,23,24,25]. Moreover, despite the large number of commercially available antihypertensive drugs, the percentage of patients with well-controlled BP is not what we would expect, and many of them either stop their therapy or follow it poorly [13,26,27].

Melatonin, N-acetyl-5-methoxytryptamine, is a ubiquitous and widely distributed indoleamine that can be found in almost all living organisms [28]. Melatonin passes through the modulation of the hypothalamic/pituitary axes ending with more complex functions [29,30]. Overall, it is considered one of the main antioxidant molecules that could work in many different situations. 

Increasingly, clinical trials have been designed to evaluate the activity of exogenous melatonin; however, the precise mechanism(s) by which it acts in hypertense patients is not completely clear, and its link with the pathogenesis of EH development remains controversial [31,32].

Moreover, most of the studies conducted up to now have been carried out for a limited period (90 days) and with a standard dosage of about 5–6 mg/die of melatonin [5,33,34]. 

We decided to carry out this randomized clinical trial in order to (1) better evaluate the connection between plasmatic oxidative status and vascular conditions in essential hypertense patients; (2) understand if the continued administration of a lower dosage of melatonin (according to the Ethics Committee, 1 mg/die is the maximum dosage that should be administered as an addition to the pharmacological anti-hypertensive therapy) for a longer period of time (1 year) could be beneficial not only clinically, but also to the structural changes involving the micro- and macro-vasculature of hypertensive patients. 

## 2. Results

No adverse effects that might be related to the additional melatonin were reported after 1 year of supplementation. 

Nine patients (five within the melatonin group and four in the control one) were unable to return for the 1-year evaluation due to COVID-19 pandemic restrictions, leaving 14 patients in the final groups (11 within the melatonin group and three in the control one). 

The demographic and clinical characteristics of the study population are reported in Table 1.

Of the 23 patients initially enrolled, 16 were randomly supplemented with melatonin, 10 were females (63%) and 6 were males (37%). The other seven patients only maintained their antihypertensive therapy; of these, three were females (42%) and four were males (57%). 

The patients' ages were between 40 years and 60 years (mean ± SD age: 53 ± 9 years for the melatonin group and 48 ± 6 years for the control group) and the mean body mass index (BMI) was 26 ± 4 Kg/m^2^ for the melatonin group and 25 ± 3 Kg/m^2^ for the control group.

During the first visit (T0), three sequential BP measurements were carried out in all patients. From these measurements, the average values obtained in the melatonin group were 133 ± 10 mmHg for the systolic blood pressure (SBP) and 83 ± 7 mmHg for the diastolic blood pressure (DBP), with a mean heart rate (HR) of 67 ± 12 bpm. For the control group, the data showed 128 ± 5 mmHg for the SBP and 84 ± 8 mmHg for the DBP with a mean HR of 71 ± 7 bpm. The groups did not significatively differ regarding the mean age and BMI (*p* = 0.089 and *p* = 0.452, respectively).

The haematological and haemodynamic variables, both for the melatonin and control group, are listed separately in Table 2 and Table 3 (at T0 and T1, respectively). 

Table 4 compares the data obtained in patients belonging to the melatonin group at T0 and after 1 year, at T1.

Considering the TAC values (expressed as CRE), comparing the starting values (T0) and the final ones (T1), it is possible to underline a different trend of these values in the melatonin group if compared to the control group. 

At T1, in the supplemented patients, there is a significant reduction (*p* = 0.041) in the TAC levels.

In this regard, there are conflicting opinions in the literature, but the data obtained confirms the hypothesis that, in a situation of altered oxidative balance, the plasma antioxidant system activates in a compensatory sense, shutting itself down when the oxidative stress is reduced [35].

Moreover, in the melatonin group, it is possible to highlight a correlation (Figure 1, Table 5) between a decrease in TAC levels and an opposite increase in the plasma concentration of melatonin at T1, in patients of the melatonin group.

There is also a change in TAC levels in the control group (Table 6), but in this case it is not significant. 

The different frequency with which endothelial damage and arterial stiffness occurred in both the melatonin and control groups in the first (T0) and second (T1) assessments is showed, respectively, in Table 2 and Table 3. Of greater interest, however, is what can be obtained from Table 4 and Table 5, where comparisons are made within each group between the first and second cardiological assessments. Regarding this, Table 4 clearly shows how, considering the values of arterial stiffness and rigidity, these improve significantly (*p* = 0.022) in patients who have taken melatonin, remaining almost unchanged, or worsening in the controls (Table 5).

Considering the endothelial dysfunction, it is possible to evaluate the Aix@75 values obtained by the EndoPAT, confirming that in this case there is also an increase in the functionality in patients supplemented with melatonin, even if it is not significant (*p* = 0.688). This point is strictly related to the mean age of the enrolled patients: considerations obtained by EndoPAT (an example from one patient is reported in Figure 2) are used as a prognostic factor on which it is possible to work preferably on the starting phases of the diseases, but even more, in patients younger than those who were enrolled in our study [36].

Ultimately, considering the SBP and the DBP over time, it is possible to see an improvement, especially in patients supplemented with melatonin. Even if, in this case, it is a non-significant improvement (*p* = 0.401 for the SBP and *p* = 1.000 for the DBP), it is interesting to note that there is a worsening of the mean values of both SBP and DBP in the control group.

## 3. Discussion

The role of oxidative stress in the pathogenesis and development of CVDs is well known, but, even if raised BP remains one of the leading causes of morbidity worldwide, only a few studies to date have focused—among all the cardiological and cardiovascular conditions—exclusively on the EH [14,37,38,39]. 

The link between antioxidants and cardiovascular physiology has a rather long history, but only recently has more space been given to the clinical meaning and relevance that this relationship may have. Specifically, among the antioxidants, special attention was given to melatonin, according to its antihypertensive, antioxidant, and vasoprotective effects [40,41]. The main point was the possibility of acting on the morphostructural alterations induced by EH, improving in the short and long term the expectation and quality of life of the patients involved [5,42,43].

Considering what was said before, this clinical trial was born with the objective of deepening the correlation between EH, oxidative stress, melatonin, and vascular alterations in humans, verifying the correlation between oxidative imbalance and the vascular damage, but, in particular, evaluating the potential protective and therapeutic role of an antioxidant supplementation in hypertensive patients. 

The development of EH can be linked to several factors: anatomical, genetic, endocrine, humoral, haemodynamic, environmental, and neural [44]. The mechanistic pathways by which EH originates have not yet been completely understood. EH onset and maintenance are both linked to endothelial dysfunction, chronic low-grade inflammation, and structural remodelling [44,45,46]. 

The healthy endothelium can be identified as a true active organ that is able to release into the circulation contractile and relax substances, growth factors, prothrombotic and antithrombotic factors, in order to promote and prevent, respectively, clot formation and anti- and pro-inflammatory mediators [44]. Further, when an imbalance in this production occurs, endothelial dysfunction follows.

The aorta and arteries play a key role in both BP and peripheral blood flow regulation affecting the effectiveness of cardiac haemodynamic, which is often compromised in the hypertensive patient [47]. In addition, vascular compliance—gradually reduced in hypertensive patients—is a key determinant of left ventricular performance; thus, alterations of the arterial system not only reflect on the peripheral component, but, over time, also compromise systemic function [47]. Overall, the entire situation is highly complex and difficult to study and manage; therefore, the literature contains discordant opinions on the effect of antioxidant therapy in EH, predominantly because only a few authors introduce pathophysiological data in the clinic [23]. 

Over the years, several pharmacological classes of anti-hypertensive drugs have been approved in clinical settings, considering that they can improve the vascular dysfunction seen in hypertension both when supplied alone or in association, but this is insufficient [44].

For this reason, it is important to remember that there is now an abundant availability of natural antioxidants. Some studies focus on vitamin D [23], others on vitamin A, and still others on vitamin E. Due to the clinical evidence for melatonin in cardiovascular health and its hypotensive effects, it is now being investigated as a non-traditional anti-hypertensive medication in patients with both essential and nocturnal hypertension, considering, first of all, its wide spectrum of action and its possibility to act both in a receptor-mediated and in a “free” mode [5]. 

To our knowledge, few studies have sought to find a correlation between oxidative stress (in our case, TAC values in plasma) and endothelial alteration associated with EH. 

The evaluations performed in this study allowed us to assume that (1) it is possible to define a direct relationship between plasmatic TAC levels and vascular alterations; in particular, arterial stiffness, (2) acting by improving plasma oxidative balance—administering an additional amount of melatonin—could lead to an enhancement in vascular functionality; and (3) considering the lack of adverse events related to the melatonin’s administration, it is possible to consider it for a future perspective, in order to include it as a part of combined treatment plans to achieve a clinical and prognostic improvement in hypertensive patients. 

In particular, our results allowed us to understand that supplementation with melatonin is beneficial, even if delivered at lower dosages than are normally used (1 mg vs. 5–6 mg). However, it must be considered that the administration time in our case was longer (1 year vs. 90 days maximum).

Despite all the precautions taken by the operators, our study has limitations that must be considered. The validity and reproducibility of estimating aortic BP, Aix@75, and PWV under ambulatory conditions by the ambulatory BP monitoring device (Mobil-O-Graph) applied in this study were confirmed in previous studies [48]. The clinical relevance of evaluating cardiac index and total vascular resistance using the same device was discussed in earlier studies; however, this device provides estimates of cardiac index and total vascular resistance that are predicted by mathematical transformations of the brachial pulse wave, and their physiological and clinical relevance require further elucidation. 

Another limitation of this clinical investigation is represented by the small number of patients involved. However, we only included consecutive newly diagnosed cases without cardiovascular risk factors to avoid selection biases, and so, in the beginning, it was difficult to reach a large number of patients who were appropriate to be studied.

Ultimately, the main limitation of the study is perhaps represented by the lack of a complete follow-up evaluation by the whole population, a bias upon which we could not act due to the tight constraints and concerns of the patients themselves associated with the COVID-19 pandemic.

## 4. Materials and Methods

### 4.1. Study Design

This study was born from a collaboration between the Cardiology Unit, ASST-Spedali Civili, Brescia, Italy and the Anatomy and Physiopathology Division, Department of Clinical and Experimental Sciences, University of Brescia, Italy. 

The study was conducted in accordance with the 1975 Declaration of Helsinki, and the protocol was approved by the Ethics Committee of the ASST-Spedali Civili, Brescia, Italy (code NP 2717, authorized Prot. n. 0042068, approved 04.07.2017). 

The investigation was designed as a randomized, prospective, and monocentric control trial with nutritional supplement, as described in Figure 3. 

The recruitment of patients was performed in the Cardiology Unit, ASST-Spedali Civili of Brescia, Italy, from March 2018 to April 2019 and comprised patients aged 40–60 years at the time of enrolment who were affected by EH and followed by the Cardiology Unit.

For the enrolment, the following were considered as exclusion criteria:-Any kind of heart disease (e.g., angina pectoris, myocardial infarction, coronary revascularization, congestive heart failure, aortic coarctation);-Autoimmune, rheumatological, or vascular diseases other than EH;-Antihypertensive therapies with nitrates, statins, or β-blockers;-Pregnancy and lactation;-Obesity;-Diabetes;-Irregular sleep/wake rhythm, such as workers with night shifts (for a period of less than 3 months before recruitment).

On the contrary, the following were considered as inclusion criteria:-A diagnosis of essential hypertension dating back at least 1 year before the start of the study;-Treatment with antihypertensive drugs already in progress and not modifiable during the course of the study;-No other cardiological (i.e., dyslipidaemia, heart failure, atrial fibrillation) and metabolic (i.e., diabetes) comorbidities;-No pregnancy in progress;-Regular sleep/wake rhythm (no worker with night shifts for a period of less than 3 months before recruitment);-Fasting blood sugar < 100 mg/dL;-Total cholesterol < 200 mg/dL and triglycerides < 150 mg/dL;

Moreover, the patients were not allowed to change their BP medication during the trial. 

Considering the above-mentioned criteria, 23 patients of either gender were enrolled (n = 23, 10 male and 13 female) after signing their informed consent. The sample size (alpha 0.05, power 80%) needed to capture 0.25 of the RH-PAT index 17, 0.5 m/s in cfPWV, was 12. The calculation was performed online using the input values of a power of 80% and a two-sided level of significance of 5% [49,50,51].

The patients were randomized 1:1 in two groups (control and supplemented group) by random number generation that minimized selection bias. The participants belonging to the supplemented group were given a dietary integration of 1 mg/die of melatonin for 1 year (“Melatonin Pura^®^”, 1 mg, 120 microtablets, linea Notte Relax, ESI srl, Albisola Superiore, SV, Italy), which was approved by the Ministry of Health of the Republic of Italy (Number: 64598) [5]. All patients underwent blood pressure measurement and a micro- and macro-vascular evaluation at the beginning of the study (T0, time of enrolment) and at the end (T1, after 1 year of daily supplementation), as described below and summarized in Table 7.

### 4.2. Blood Pressure Measurement

Blood pressure was assessed using a standard, calibrated sphygmomanometer. The mean of three sitting and standing blood pressure results, respectively, was recorded. The arm in which the highest sitting DBP was found was the arm used for all subsequent readings throughout the study. Every effort was made to have the same staff member obtain blood pressure measurements in each individual patient, at the same time of day, using the same equipment. SBP was recorded when the initial sound was heard (Phase I of the Korotkoff sound), while DBP was noted when it ceased (Phase V of the Korotkoff sound). The cuff was deflated at a rate not greater than 2 mmHg/s [52]. 

### 4.3. Circulating Total Antioxidant Capacity Levels

Peripheral venous blood samples were carried out between 2 and 4 p.m. when melatonin is not physiologically secreted, according to Cagnacci and colleagues' indications [53]. The blood samples were collected in ethylenediamine tetraacetic acid (EDTA) test tubes; the plasma was obtained by centrifugation (3000 rpm, 15 min at 4 °C) within a maximum of 2 h after extraction, immediately aliquoted, and stored at −80 °C until use [54]. 

The plasmatic antioxidant capacity was assayed using a colorimetric assay kit (MyBioSource, Inc., San Diego, CA, USA) that measures Cupper (Cu)++ reduction to Cu+ by the antioxidant factors in the sample, by coupling with a colorimetric probe. After processing as described by the manufacturer’s instructions, the microplate was read at 490 nm (Sunrise, Tecan; Männedorf, Switzerland), and the plasmatic TAC was expressed as μM Copper Reducing Equivalents (CRE) [35]. 

Currently, the value of total antioxidant capacity appears to be a valid indicator in the context of alimentary supplementation [55,56]. In addition, the TAC value allows us to find a correlation with dietary TAC (DTAC is considered more in recent studies), thus, associating the patient’s nutritional status (BMI) with the vascular damage possibly associated with the change in oxidative stress [56].

### 4.4. Cardiovascular Evaluation

#### 4.4.1. Peripheral Arterial Tonometry: Assessment of the Endothelial Function 

Peripheral arterial tonometry (PAT) was obtained using EndoPAT-2000 (Itamar Medical Ltd., Caesarea, Israel), which was previously validated in numerous populations [50,51]. The examination consists of a protocol of reactive ischemia of the upper limb with plethysmographic measurement of the pulse amplitude by means of disposable pneumatic probes, which are applied to the hand indexes and inflated at DBP to perceive the oscillations of the arterial pulse, avoiding the interference of the venous-arteriolar reflex. The patient lies supine in a temperature-controlled environment and the measurement lasts 15 min, of which the 5 central ones are characterized by the ischemia of a limb obtained by inflating the sleeve of a sphygmomanometer at suprasystolic pressure, and the 5 initial and 5 final ones with a deflated sleeve. The software elaborates a reactive hyperaemia index (RHI) comparing the increase in blood flow in the ischemized limb with the baseline and the contralateral limb.

The examination in this study involved the use of specially designed finger probes placed on the middle finger of each subject’s hands. As showed in Figure 4, endothelial function was measured via a reactive hyperaemia (RH) protocol consisting of a 5 min baseline measurement, after which a BP cuff on the test arm was inflated to 60 mmHg above baseline SBP or to at least 200 mmHg for 5 min. Occlusion of the pulsatile arterial flow was confirmed by the reduction in the PAT tracing to zero. After 5 min, the cuff was deflated, and the PAT tracing was recorded for a further 5 min.

The ratio of the PAT signal after cuff release compared with baseline was calculated through a computer algorithm, automatically normalizing for baseline signal and indexed to the contralateral arm. The calculated ratio is called the RH-PAT index or RHI. Its normal value is >2.00 (pure number), while a clear endothelial dysfunction is shown by a value ≤ 1.67. The software calculated the peripheral augmentation index (AIx), which is a measure of arterial stiffness. In particular, since this parameter is influenced by HR, it was standardized automatically per 75 bpm (AIx@75). According to EndoPAT software, peripheral AIx@75 is normal when <17%.

#### 4.4.2. SphygmoCor System Evaluation: Study of the Pulse Wave Velocity

The central aortic haemodynamic parameters, the augmentation index at a heart rate of 75 beats per minute (AI@75), the augmentation pressure (AP), and the pulse pressure (PP) were measured using the applanation tonometry method and the SphygmoCor system (Atcor Medical, Sydney, Australia) [57]. All measurements were performed by the same operator. 

The tip of the tonometer was pressed gently against the radial artery at the site of maximum pulsation at the wrist. This micromanometer precisely records pressure within the artery, recording the pulse wave for 10–12 s (Millar Instruments, Houston, TX, USA) [58]. 

A generalized transfer function [59] was applied to the radial artery waveform to derive the aortic waveform. Central SBP and DBP, aortic pulse pressure (PP_ao_), and left ventricular end-systolic pressure (LVESP) were determined with the integrated software. AIx is a measure of the stiffness of the arterial walls. As there is a linear relationship between this and the heart rate (HR), AIx was standardized to an HR of 75 bpm (AIx@75).

The AP is defined as the height of the late systolic peak above the inflection point; in healthy patients of cohort sizes between 25 and 88, the within-observer difference varied from 0.5 ± 5.4 to 1.4 ± 1.2%. 

In the same way, carotid and femoral waveforms were recorded and the distance between the two points measured. By means of electrocardiogram (ECG) gating, the ratio between this distance to the travelling time of the pulse wave (using the foot–foot convention) is called the carotid-femoral pulse wave velocity (cfPWV), a measure of arterial stiffness (Figure 5 and Figure 6). 

### 4.5. Follow Up Evaluation

At the one-year follow up, all participants were invited for a second medical examination, in which all the evaluations were performed (both the venous sampling for the laboratory’s tests and the clinical analysis for the cardiological assessment).

### 4.6. Statistical Analysis

The statistical analysis of the various parameters examined was carried out after possible transformation of the non-normal distribution variables, by means of a two-way (time-treatment) variance analysis. Bivariate correlations between the variation of blood parameters and cardiovascular parameters were carried out, and a value of *p* < 0.05 was considered significant. Each statistical test was a two-way test.

As regards the number of patients, reference is made to recent studies carried out in the Cardiology Unit of ASST-Spedali Civili, Brescia, Italy [53]. The sample size (alpha 0.05, power 80%) needed to capture 0.25 of the RH-PAT index 17, 0.5 m/s in cfPWV, is 12.

## 5. Conclusions

EH is a wide and complex pathology, and it is impossible to consider that it might be solved with the administration of a single drug. Moreover, the drugs that are currently used have a pool of side effects that must be taken into account, and that often require the introduction of blended therapies. Moreover, not all patients are able to tolerate the drug combinations that are often proposed to reduce these side effects. In this sense, identifying a supplementation that can be added to the current main therapy’s schemes without affecting the patient’s balance would be a considerable step forward, particularly considering the data confirming the presence of a very small number of side effects attributable to melatonin.

Despite all the limitations to bear in mind, we believe that these results may represent a good pathophysiological basis for future studies that aim to better analyse a correlation between hypertensive patients of different ages and sex, being able to involve a larger number of patients that could be studied prospectively for several years in order to highlight an effective clinical outcome, and giving more importance to the new non-invasive methods that could represent a real breakthrough in the prevention of major cardiovascular events.

## Figures and Tables

**Figure 1 ijms-23-14489-f001:**
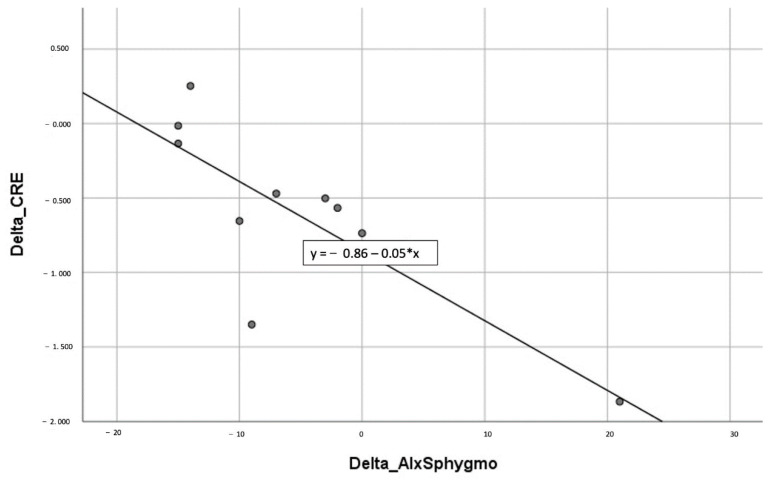
Schematic representation of the relationship between applanation tonometry performed by the SphygmoCor system and the TAC plasmatic values. (CRE: μM Copper Reducing Equivalents; Aix: augmentation index).

**Figure 2 ijms-23-14489-f002:**
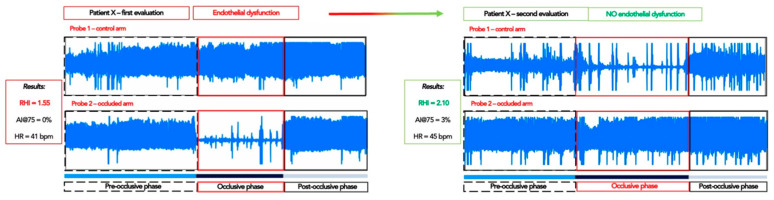
Schematic representation obtained from the EndoPAT analysis: comparison between the first assessment (T0) and the one-year assessment (T1). (RHI: reactive hyperaemia index; Aix: augmentation index; Aix@75: augmentation index corrected for 75 bpm; HR: heart rate).

**Figure 3 ijms-23-14489-f003:**
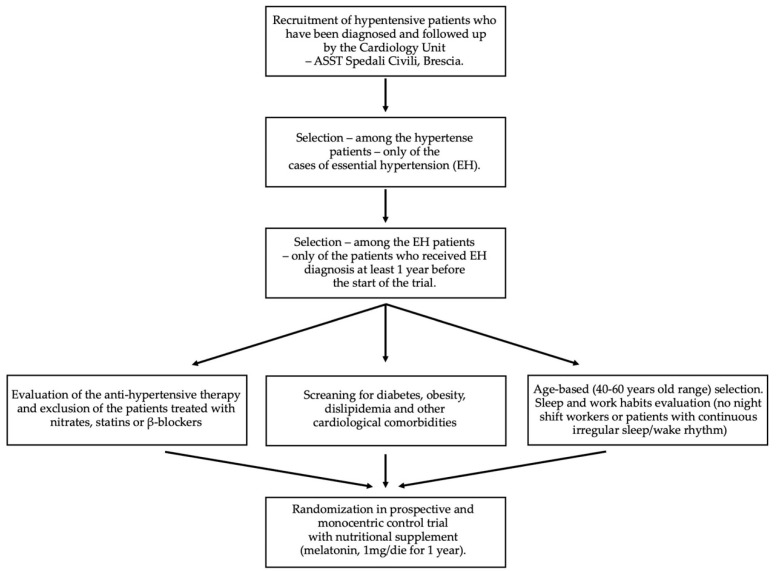
Schematic representation of the study design.

**Figure 4 ijms-23-14489-f004:**
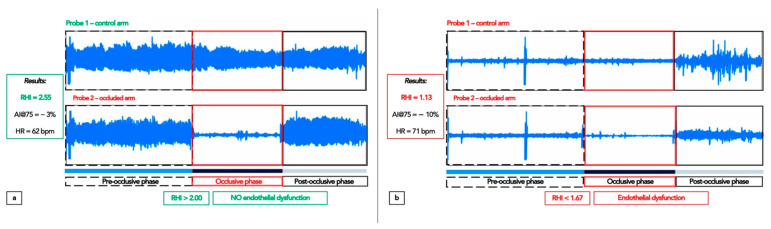
Peripheral arterial tonometry (PAT) signals obtained using the EndoPAT-2000 device, showing traces typical of good endothelial function (**a**) and of endothelial dysfunction (**b**). Note the difference in the post-occlusive phase, with lower amplitude, similar to baseline trace, in the latter.

**Figure 5 ijms-23-14489-f005:**
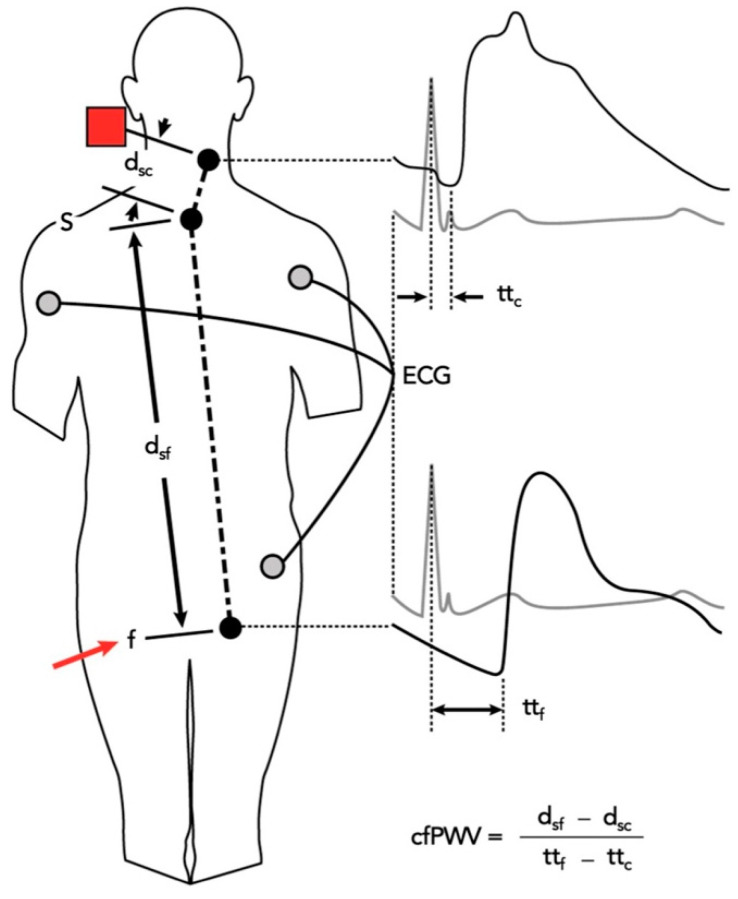
Schematic representation of SphygmoCor CvMS carotid-femoral pulse wave velocity (cf- PWV) measurement method with sequential applanation tonometry of the carotid (marked as a red square) and femoral (marked as a red arrow) sites. Pulse transit times (tt) are calculated from the electrocardiogram (ECG) R-wave to the foot of the applanated waves. Distances (d) are measured from the suprasternal notch (s) to the sites of applanation tonometry. Modified from Butlin and Qasem, 2016; © 2016 by S. Karger AG, Basel [60].

**Figure 6 ijms-23-14489-f006:**
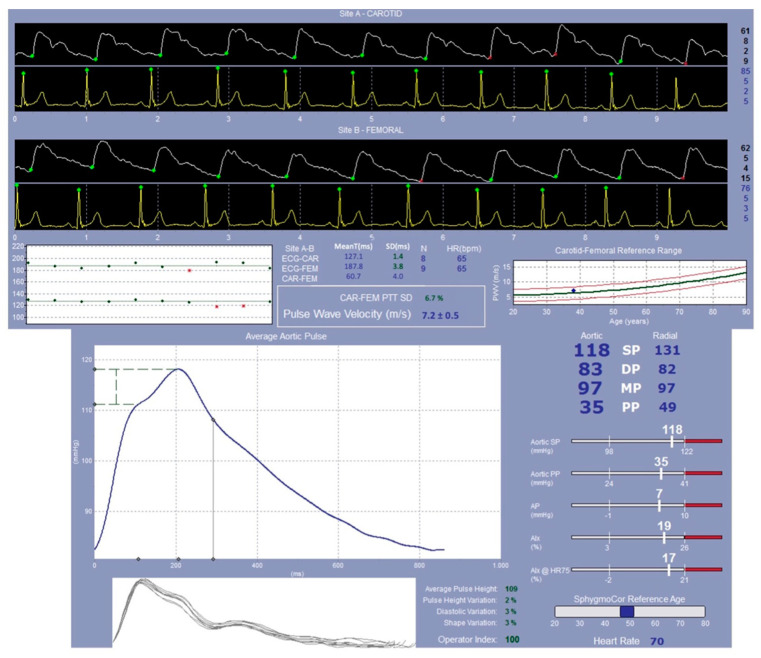
Schematic representation of the data obtained from applanation tonometry performed by the SphygmoCor system (Atcor Medical, Sydney, Australia).

**Table 1 ijms-23-14489-t001:** Demographic and clinical description of study cohort of patients with essential hypertension (HE) obtained during the first evaluation (T0). Data are expressed as mean ± standard deviation (SD). (BMI: body mass index; SBP: systolic blood pressure; DBP: diastolic blood pressure; HR: heart rate).

Characteristics	Melatonin T0 (*n* = 16)	Control T0 (*n* = 7)	Intergroup ANOVA (*p*)
Age at the time of recruitment—T0 (years)	53 ± 9	48 ± 6	0.089
Sex	F: 10 (63%) M: 6 (37%)	F: 3 (42%) M: 4 (77%)	0.0650
BMI (Kg/m^2^)	26 ± 4	25 ± 3	0.452
SBP (mmHg)	133 ± 10	128 ± 5	0.278
DBP (mmHg)	83 ± 7	84 ± 8	0.820
HR (bpm)	67 ± 12	71 ± 7	0.680

**Table 2 ijms-23-14489-t002:** Baseline evaluation (T0) of total antioxidant capacity (TAC), endothelial function with non-invasive method (EndoPAT), and peripheral plethysmography for the evaluation of indices of arterial stiffness (SphygmoCor) in both melatonin and control group. Data are expressed as mean ± standard deviation (SD).

Variable	Melatonin T0 (*n* = 15)	Control T0 (*n* = 7)	Intergroup ANOVA (*p*)
TAC (CRE)	1.49 ± 0.53	1.40 ± 0.66	1.000
RHI RHI ≤ 1.67 RHI ≤ 2.00	1.86 ± 0.71 7 (46.7%) 8 (53.3%)	2.00 ± 0.58 1 (14.3%) 3 (42.9%)	0.731 0.193 1.000
Aix@75 (%)	12.33 ± 16.38	9.43 ± 17.98	0.731
Aix@75 ≥ 17%	6 (40.0%)	2 (28.6%)	1.000
SBP (mmHg)	133 ± 10	128 ± 5	0.278
DBP (mmHg)	83 ± 7	84 ± 8	0.820
MAP (mmHg)	104 ± 9	101 ± 8	0.308
PP (mmHg)	49 ± 6	44 ± 6	0.154
HR (bpm)	67 ± 12	71 ± 7	0.680
SBP_ao_ (mmHg)	127 ± 12	120 ± 6	0.175
DBP_ao_ (mmHg)	90 ± 12	86 ± 8	0.413
PP_ao_ (mmHg)	36 ± 7	36 ± 5	0.802
AP_ao_ (mmHg)	10 ± 8	12 ± 5	0.680
Aix@75 (%)	26.08 ± 19.84	23.67 ± 11.23	0.368
Aix@75 ≥ 35%	7 (50.00%)	1 (16.7%)	0.325
cfPWV (m/s)	5.36 ± 2.65	8.91 ± 10.13	0.588
cfPWV > 9.6 m/s	0 (0.00%)	1 (14.7%)	0.350

CRE: μM Copper Reducing Equivalents; RHI: reactive hyperaemia index; Aix: augmentation index; Aix@75: augmentation index corrected for 75 bpm; SBP: systolic blood pressure; DBP: diastolic blood pressure; MAP: mean arterial pressure; PP: pulse pressure; HR: heart rate; SBP_ao_: aortic systolic blood pressure; DBP_ao_: aortic diastolic blood pressure; PP_ao_: aortic pulse pressure; AP_ao_: aortic augmentation pressure; cfPWV: carotid-femoral pulse wave velocity.

**Table 3 ijms-23-14489-t003:** Final evaluation (T1) of plasma parameters' total antioxidant capacity (TAC), endothelial function with non-invasive method (EndoPAT), and peripheral plethysmography for the evaluation of indices of arterial stiffness (SphygmoCor) in both melatonin and control group. * Standard deviation not reported; only one value analysed. Data are expressed as mean ± standard deviation (SD).

Variable	Melatonin T1 (*n* = 9)	Control T1 (*n* = 6)	Intergroup ANOVA (*p*)
TAC (CRE)	0.92 ± 0.39	1.50 ± 0.69	0.885
RHI RHI ≤ 1.67 RHI ≤ 2.00	1.54 ± 0.67 6 (60.0%) 6 (60.0%)	1.81 ± 0.32 1 (33.3%) 2 (66.7%)	0.573 0.559 1.000
Aix@75 (%)	5.30 ± 11.09	−4.67 ± 21.50	0.469
Aix@75 ≥ 17%	2 (20.0%)	1 (33.3%)	1.000
SBP (mmHg)	129 ± 14	133 ± 22	0.600
DBP (mmHg)	86 ± 6	92 ± 2	0.145
MAP (mmHg)	102 ± 9	101 ± 16	0.921
PP (mmHg)	42 ± 10	47 ± 11	0.497
HR (bpm)	71 ± 13	62 ± 9	0.373
SBP_ao_ (mmHg)	120 ± 14	121 ± 19	0.776
DBP_ao_ (mmHg)	87 ± 7	86 ± 11	1.000
PP_ao_ (mmHg)	33 ± 9	35 ± 8	0.497
AP_ao_ (mmHg)	8 ± 5	8 ± 1	0.630
Aix@75 (%)	23.88 ± 10.02	16.67 ± 5.03	0.133
Aix@75 ≥ 35%	-	-	-
cfPWV (m/s)	6.20 ± 2.95	1.8 *	0.500
cfPWV > 9.6 m/s	-	-	-

CRE: μM Copper Reducing Equivalents; RHI: reactive hyperaemia index; Aix: augmentation index; Aix@75: augmentation index corrected for 75 bpm; SBP: systolic blood pressure; DBP: diastolic blood pressure; MAP: mean arterial pressure; PP: pulse pressure; HR: heart rate; SBP_ao_: aortic systolic blood pressure; DBP_ao_: aortic diastolic blood pressure; PP_ao_: aortic pulse pressure; AP_ao_: aortic augmentation pressure; cfPWV: carotid-femoral pulse wave velocity.

**Table 4 ijms-23-14489-t004:** Schematic representation of the comparison of the different variables studied, within the melatonin group, at T0 and T1. ^#^
*p* < 0.05 vs. Melatonin T0 group. Data are expressed as mean ± standard deviation (SD).

Variable	Melatonin T0	Melatonin T1	Intergroup ANOVA (*p*)
TAC (CRE)	1.41 ± 0.61	0.92 ± 0.39	0.041
RHI RHI ≤ 1.67 RHI ≤ 2.00	1.97 ± 0.74 7 (46.7%) 8 (53.3%)	1.54 ± 0.67 6 (60.0%) 6 (69.0%)	0.241 0.688 1.000
Aix@75 (%)	12.40 ± 18.43	5.30 ± 11.09	0.020
Aix@75 ≥ 17%	6 (40.0%)	2 (20.0%)	0.402
SBP (mmHg)	135 ± 10	129 ± 14	0.401
DBP (mmHg)	86 ± 6	86 ± 6	1.000
MAP (mmHg)	104 ± 9	102 ± 9	0.779
PP (mmHg)	48 ± 10	42 ± 10	0.204
HR (bpm)	73 ± 11	71 ± 13	0.944
SBP_ao_ (mmHg)	125 ± 9	120 ± 14	0.484
DBP_ao_ (mmHg)	89 ± 11	87 ± 7	0.799
PP_ao_ (mmHg)	35 ± 8	33 ± 9	0.674
AP_ao_ (mmHg)	9 ± 8	9 ± 5	1.000
Aix@75 (%)	26.57 ± 21.07	23.43 ± 10.73	0.345
Aix@75 ≥ 35%	7 (50.0%)	0 (0.0%)	0.022 ^#^
cfPWV (m/s)	4.60 ± 2.90	6.33 ± 3.20	0.345
cfPWV > 9.6 m/s	-	-	-

CRE: μM Copper Reducing Equivalents; RHI: reactive hyperaemia index; Aix: augmentation index; Aix@75: augmentation index corrected for 75 bpm; SBP: systolic blood pressure; DBP: diastolic blood pressure; MAP: mean arterial pressure; PP: pulse pressure; HR: heart rate; SBP_ao_: aortic systolic blood pressure; DBP_ao_: aortic diastolic blood pressure; PP_ao_: aortic pulse pressure; AP_ao_: aortic augmentation pressure; cfPWV: carotid-femoral pulse wave velocity.

**Table 5 ijms-23-14489-t005:** Schematic representation of the comparison of the different variables studied, within the control group, at T0 and T1. ^#^ Standard deviation not reported; only one value analysed. Data are expressed as mean ± standard deviation (SD).

Variable	Control T0	Control T1	Intergroup ANOVA (*p*)
TAC (CRE)	1.63 ± 0.34	1.50 ± 0.68	0.539
RHI RHI ≤ 1.67 RHI ≤ 2.00	2.19 ± 0.30 1 (14.3%) 3 (42.0%)	1.81 ± 0.32 1 (33.3%) 2 (66.7%)	0.109 1.000 1.000
Aix@75 (%)	−1.33 ± 17.55	−4.67 ± 21.50	0.180
Aix@75 ≥ 17%	2 (28.6%)	1 (33.3%)	1.000
SBP (mmHg)	131 ± 8	133 ± 2	0.655
DBP (mmHg)	88 ± 10	92 ± 2	0.593
MAP (mmHg)	105 ± 13	101 ± 16	1.000
PP (mmHg)	43 ± 2	47 ± 11	0.414
HR (bpm)	68 ± 7	62 ± 9	0.109
SBP_ao_ (mmHg)	122 ± 10	121 ± 9	1.000
DBP_ao_ (mmHg)	89 ± 11	86 ± 11	1.000
PP_ao_ (mmHg)	32 ± 2	35 ± 8	0.414
AP_ao_ (mmHg)	10 ± 2	8 ± 1	0.414
Aix@75 (%)	27.33 ± 9.07	16.67 ± 5.03	0.109
Aix@75 ≥ 35%	1 (16.7%)	0 (0.0%)	1.000
cfPWV (m/s)	6.90 ^#^	1.80 ^#^	0.317
cfPWV > 9.6 m/s	1 (14.3%)	0 (0.0%)	1.000

CRE: μM Copper Reducing Equivalents; RHI: reactive hyperaemia index; Aix: augmentation index; Aix@75: augmentation index corrected for 75 bpm; SBP: systolic blood pressure; DBP: diastolic blood pressure; MAP: mean arterial pressure; PP: pulse pressure; HR: heart rate; SBP_ao_: aortic systolic blood pressure; DBP_ao_: aortic diastolic blood pressure; PP_ao_: aortic pulse pressure; AP_ao_: aortic augmentation pressure; cfPWV: carotid-femoral pulse wave velocity.

**Table 6 ijms-23-14489-t006:** Non-parametric correlation between vascular indexes and oxidative markers. Data are expressed as mean ± standard deviation (SD).

Variable	RHO Correlation	Intergroup ANOVA (*p*)
ΔRHI/ΔAix EndoPAT	0.416	0.204
ΔRHI/ΔAix SphygmoCor	0.299	0.471
ΔRHI/ΔPWV	0.800	0.200
ΔRHI/ΔTAC (CRE)	0.442	0.174
ΔAix EndoPAT/ΔAix SphygmoCor	−0.571	0.139
ΔAix EndoPAT/ΔPWV	−0.800	0.200
ΔAix EndoPAT/ΔTAC (CRE)	0.483	0.132
ΔAix EndoPAT/ΔMelatonin	0.232	0.658
ΔAix SphygmoCor/ΔPWV	−0.158	0.663
ΔAix SphygmoCor/ΔTAC (CRE)	−0.729	0.017

CRE: μM Copper Reducing Equivalents; RHI: reactive hyperaemia index; Aix: augmentation index; Aix@75: augmentation index corrected for 75 bpm; SBP: systolic blood pressure; DBP: diastolic blood pressure; MAP: mean arterial pressure; PP: pulse pressure; HR: heart rate; SBP_ao_: aortic systolic blood pressure; DBP_ao_: aortic diastolic blood pressure; PP_ao_: aortic pulse pressure; AP_ao_: aortic augmentation pressure; cfPWV: carotid-femoral pulse wave velocity.

**Table 7 ijms-23-14489-t007:** Schematic summarizing of the biomarkers that were considered and of the cardiological evaluations that were carried out.

Name of the Procedure	Laboratory Test/ Cardiological Evaluation	Type of Sample	Time Point
Blood pressure measurement	Cardiological evaluation	N/A	T0 and T1
Total antioxidant capacity level	Laboratory test (ELISA test)	Plasma	T0 and T1
Peripheral arterial tonometry—endothelial function	Cardiological evaluation (EndoPAT-2000)	N/A	T0 and T1
Pulse wave velocity	Cardiological evaluation (SphygmoCor system)	N/A	T0 and T1

## Data Availability

Not applicable.

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
