# Peer review of "Essential Hypertension and Oxidative Stress: Novel Future Perspectives"

_ijms, 2022, doi:10.3390/ijms232214489_

Round 1

Reviewer 1 Report

This is a very interesting study with unique methodology. The authors found melatonin supplementation improves arterial stiffness significantly. Moreover, the supplemented group conveyed a significative reduction in total antioxidant capacity levels that was correlated with the improvement of the arterial stiffness. I have a few questions:

-How did the authors define a normal circadian/sleep cycle?

-Were any of the patients taking other anti-oxidants such as N-acetylcysteine?

-Do you plan to explore the role of melatonin in patients with established cardiovascular disease/atherosclerosis?

-Minor spell and grammar edits would be useful

Author Response

We thank the Referee for all the indications.

- How did the authors define a normal circadian/sleep cycle? We thank the Referee for the question. Actually, a specific definition of circadian/sleep cycle would require a series of investigations. Recent studies, for example, use tools such as Actigraphy for monitoring the activities and scoring the sleep-wake rhythm (Nakashima et al., 2022). In our case, however, the category "regular sleep/wake rhythm" and the category "irregular sleep/wake rhythm" in the inclusion and exclusion criteria simply allowed us to exclude patients who had persistently a reversal sleep-wake rhythm or non-constant rhythm, thus excluding night workers or shift workers, without considering more specific variables regarding overall sleep quality. We thank the Referee for this clarification: we rewrite the exclusion criteria in order to better specify the meaning of the restriction.

- Were any of the patients taking other anti-oxidants such as N-acetylcysteine? We thank the referee for the question. No, none of the other patients were doing other types of supplementations during the study trial.

Do you plan to explore the role of melatonin in patients with established cardiovascular disease/atherosclerosis? We thank the Referee for the question. These are the first data we collected. The project initially also included an echocardiographic evaluation of the patients with the possibility of later reevaluating them also in the long term to see how long-term the impact of melatonin administration could be. The data collected, however, are still being processed. It would also be interesting to conduct a prospective study with more prolonged administration of melatonin over time. On the other hand, considering established cardiovascular disease and atherosclerosis, we believe that it could be a fruitful background for future studies. There are more and more research groups taking an interest in the topic, delving into different themes. Many are focusing on the positive role melatonin may have on mitochondrial function for example (Guo et al., 2022; Chen et al., 2022), but also on the systemic effect it may have (Tobeiha et al., 2022; Fernández-Ortiz et al., 2022; Guo et al., 2022). It will certainly be a future target to invest in.

-Minor spell and grammar edits would be useful. We thank the Referee for the indication. The text has been proofread and corrected (all the corrections have been added in red in the text).

Reviewer 2 Report

The study evaluates endothelial damage and the vascular stiffness in hypertensive patients and tests the effect of the supplementation with physiological antioxidant melatonin in patients with essential hypertension.

The authors have already mentioned the main limitations of the study, including small number of patients, using device that evaluates cardiac index and total vascular resistance by mathematical transformations, lack of complete follow-up evaluation by the whole population.

Apart from this, can you explain why did you measure only total antioxidant capacity (TAC), as a parameter of oxidative stress? TAC may be reduced because oxidative stress is decreased, but also a possible explanation may be that oxidative stress is increased and TAC is reduced because it is spent to neutralize reactive oxygen species. The results would be more valid if some additional parameters of oxidative stress are measured, such as TBARS, MDA....and compared together with TCA.

The full name of parameters abbrevations are listed in the section Material and Methods, but it will be easier to follow the results, if they are incorporated also in the Table legend, when they are mentioned for the first time.

Some English corrections are also needed.

Author Response

We thank the Referee for all the indications.

- can you explain why did you measure only total antioxidant capacity (TAC), as a parameter of oxidative stress? TAC may be reduced because oxidative stress is decreased, but also a possible explanation may be that oxidative stress is increased and TAC is reduced because it is spent to neutralize reactive oxygen species. The results would be more valid if some additional parameters of oxidative stress are measured, such as TBARS, MDA....and compared together with TCA. We thank the Referee for the consideration. Many of the early studies involving the melatonin supplementation and the cardiovascular diseases focused on assessing plasma oxidative status by quantifying total antioxidant capacity (TAC) (Reiter et al., 2005; Mistraletti et al., 2017). More recent projects, however, have been open to the use of novel markers of oxidative stress and redox balance, but in fact, initially circulating TAC values appeared to be among the most reliable for verifying a change in oxidative stress at the plasma level (Bouroutzika et al., 2022). For this reason, the ethics committee initially approved conducting only one test that could be as representative as possible of what they wanted to study. At the moment, in any case, the value of total antioxidant capacity appears to be a very valid indicator in the context of alimentary supplementation (Galiñanes et al., 2022; Zujko et al., 2022). In addition, the TAC value also allows us to find a correlation with dietary TAC (- DTAC, more considered in recent studies), thus also associating the patient's nutritional status (BMI) with the vascular damage possibly associated with the change in oxidative stress (Zujko et al., 2022). We thank the Refere for the clarification: we have added a short comment in the part of Material and Methods section in which oxidative stress evaluation is assessed.

- The full name of parameters abbrevations are listed in the section Material and Methods, but it will be easier to follow the results, if they are incorporated also in the Table legend, when they are mentioned for the first time. We thank the Referee for the suggestions. We change the abbreviations in the text by including them also in the Results part.

- Some English corrections are also needed. We thank the referee for the indication. The text has been proofread and corrected (all the corrections have been added in red in the text).

Round 2

Reviewer 2 Report

The authors responded to all comments and questions.